# Composition Design and Performance Evaluation of Emulsified Asphalt Cold Recycled Mixtures

**DOI:** 10.3390/ma12172682

**Published:** 2019-08-22

**Authors:** Yuhui Pi, Zhe Huang, Yingxing Pi, Guangcan Li, Yan Li

**Affiliations:** 1School of Civil Engineering and Architecture, Chongqing Vocational Institute of Engineering, Chongqing 402260, China; 2School of Highway, Chang’an University, Xi’an 710064, China

**Keywords:** cold recycled asphalt mixtures, emulsified asphalt, composition design, technical performance, fatigue characteristics

## Abstract

Based on an analysis of the cold regeneration mechanism of emulsified asphalt, the emulsified asphalt binders and cement were applied to prepare the cold recycled mixtures, and the main technical performances of the designed mixtures were evaluated, including high-temperature stability, water stability, and fatigue characteristics. A high content of 65% recycled asphalt pavement (RAP) material was used with some new aggregates and mineral powders, and the optimal emulsified asphalt binder and cement dosages were determined as 2.9% and 1.5% respectively. The technical performance test results show that: (1) The well-designed emulsified asphalt cold recycled mixtures have good high-temperature stability and water stability, and can meet the requirements of the road base layer and the lower layer. (2) When the stress level is lower, the fatigue performance of mixtures with lower emulsified asphalt binder dosage and lower cement content is better, but when the stress level is higher, the high dosage of emulsified asphalt binder is more favorable, while the cement content has little effect on the fatigue property. (3) The emulsified asphalt cold recycled mixtures have relatively poor fatigue resistance, and their fatigue life is significantly lower than that of the hot mixed asphalt mixtures.

## 1. Introduction

Along with the burgeoning construction of transportation infrastructures, the mileage of highways is constantly increasing; many of the early built highways have served for a long time, and highway performance obviously declines, bringing about the need for continuous maintenance and renovation. In recent years, highway repair and renovation have become increasingly arduous, and more and more used pavement materials are generated during those processes, such as asphalt pavement materials, cement concrete pavement materials, and so on. If the used pavement materials are treated as waste, it is not only a waste of resources, but also a use of landfill space, as well as a cause of environmental disruption. Therefore, applying recycling technologies to reuse recycled pavement materials will be a very effective measure, especially for recycled asphalt pavement (RAP) materials [1,2,3,4]. Among recycling technologies, cold recycling technology is the main way for asphalt pavement regeneration, and the use of emulsified asphalt binder for cold regeneration has rapidly become popular [5,6]. The application of emulsified asphalt cold recycled mixtures (EACRM) to pave the surface and base layers of roads can reduce construction costs, energy consumption, and environmental pollution [7]. To build a resource-saving and environment-friendly society, cold recycling technology can make full use of existing pavement materials and improve some performance defects of semi-rigid base layers, which will surely receive more and more attention. 

Due to the low strength and poor road performance, it is required to improve the material composition of EACRM so as to guarantee the durability of the pavements. The mechanism of EACRM is analyzed from a macro aspect, and the relevant factors affecting the technical performance are pointed out in [8,9,10,11]. The technical performance of EACRM designed with different RAP contents, cement contents, and gradation was evaluated according to high-temperature stability (Rutting test), low-temperature crack resistance (Low-temperature bending test), and water stability (Freeze-thaw split test and Immersion Marshall test) [12,13,14]. Creep and fatigue tests were used to evaluate the fatigue characteristics of EACRM by controlling the stress or strain levels, and corresponding fatigue failure characteristics and general rules were obtained [15,16,17,18,19]. Some scholars have analyzed the mechanism of demulsification strength of cement in EACRM from the microscopic point of view, and studied the micro-structure of mortar-aggregate interfaces and the micro-structure characteristics of simulated emulsified asphalt cement concrete [19,20].

The above studies have undoubtedly had positive influences on the improvement of the performance of EACRM, but they mainly focus on the effects of RAP content and properties, adhesion and spalling mechanism, asphalt type, additives type, and content on the pavement performance of cold recycled mixtures, and little study has been undertaken on the effects of material properties and composition ratios on the technical performance of EACRM. In addition, all the above studies used the heavy indoor compaction and Marshall methods to prepare specimens, but these molding methods cannot effectively simulate the actually filed construction, and the correlation between indoor specimens and pavement core samples is less than 70% [21]. In view of this, this study adopts the vibration compaction molding method, which is more in line with the actual situation to study the influence of material properties and composition ratio on emulsified asphalt cold recycled mixtures. The research results are of referential value in engineering practice.

## 2. Materials and Experiments

### 2.1. Raw Materials

(1) Preparation of emulsified asphalt

The emulsifier is an important component of emulsified asphalt, and plays a key role in the emulsification of asphalt [22]. The relevant performance test results show that high quality asphalt will give better technical performance, and that aggregate gradation, aggregate temperature, additional water content, cement content, and sand content significantly affect the mixing time and demulsification time [23,24,25,26,27]. Since the emulsified asphalt of cold recycled material needs sufficient working time after being mixed with stone, and considering the alkaline environment of cement hydration and the requirement of initial setting time, as well as ensuring that asphalt can participate in the formation of cement paste structure after demulsification, in this paper, a slow-cracking, slow-setting emulsifier was used. In this study, E-type emulsifier and No. 70 matrix asphalt were used to prepare emulsified asphalt. The main chemical components of the selected emulsifiers included the tallow alkyl diamine ethoxylate, nonylphenol ethoxylate, amino lignin, 2-propanol, and oil diamine ethoxylate. The technical specifications of 70 # matrix asphalt all met the performance requirements. The emulsification process adopted in this paper was as follows: The first step was the preparation of the bitumen to ensure that its temperature in the emulsification equipment was controlled at around 130 °C. The second step was the preparation of the soap liquid, selecting the appropriate emulsifier type and dosage, and the type and dosage of the additive to prepare an aqueous emulsifier solution (soap liquid). The last step was to add bitumen and soap liquid into the emulsifier; then, through the mechanical action of pressurization, shearing and grinding, the bitumen could form uniform and fine particles, which could be dispersed stably and evenly in the soap liquid to form an oil-in-water asphalt emulsion. The test indexes of emulsified asphalt are shown in Table 1.

(2) Cement

Considering the performance requirements of the pavement base, ordinary Portland cement (P.O. 32.5) was used. The test results of the cement are shown in Table 2.

(3) Mixture gradation

Yang et al studied the effects of RAP materials on the performances of cement stabilized recycled mixture, and indicated that optimal aggregates gradation should be designed before the determination of optimal contents of emulsified asphalt binder and cement [28]. So, in this study, the vibration compaction method was applied to determine the aggregate gradation as 65% RAP materials, 13% new aggregates (0–3 mm), 20% new aggregates (10–20 mm), and 2% mineral powders. The actual gradation results are shown in Table 3, which met the requirements of the gradation range of medium size in “Technical Specification for Asphalt Pavement Regeneration of Highway” JTG F41-2008. The gradation curve of cold recycled emulsified asphalt mixtures is shown in Figure 1.

### 2.2. Design of Emulsified Asphalt Cold Recycled Mixtures

The interaction between asphalt and filler plays a key role in the performance of asphalt mixtures, and is influenced by material properties [29,30]. The amount of cement used in this study was 1.0%, 1.5%, and 2.0% of the total mass of RAP and new aggregates. The amount of emulsified asphalt was set as 2.2%, 2.9%, 3.6%, and 4.3% of the mass of total aggregate [31,32]. The vibratory compaction method was used to design the mix ratio, and the optimum mix ratio was determined by the Marshall method. The Marshall stability variation with the emulsified asphalt dosage are shown in Figure 2. Variations of the void volume with different asphalt content are shown in Figure 3.

The test results showed that the void volume of three kinds of specimens with different cement contents decreased with an increase of emulsified asphalt. China’s current specification, “Technical Specification for Highway Asphalt Pavement Regeneration” (JTG F41-2008), requires that the void ratio of emulsified asphalt cold recycled mixtures be 9–14%; the recommended void ratio is less than 12%. When the content of emulsified asphalt is 2.5–2.9%, Marshall stability is the best, so 2.9% emulsified asphalt content is optimal. In this case, 2.9% emulsified asphalt and 1.5% or 2.0% cement cold recycled mixtures meet the technical requirements. In addition, adding a certain amount of cement can improve the performance of cold recycled mixtures, but excessive cement will lead to the deterioration of water stability and shrinkage characteristics of mixtures. Therefore, it is suggested that the cement content should be less than 1.5% in China’s specifications, so emulsified asphalt 2.9% and cement 1.5% are optimal. The splitting test results of cold recycled emulsified asphalt mixtures are shown in Table 4.

Gross volume density can reflect the compaction degree of the mixture specimen, and the degree of compaction affects water stability, strength, etc., to a certain extent [33]. Therefore, it was necessary to measure the gross volume density as a reference for the performance of the mixture. The gross volume density test was carried out with the cement content of RAP and aggregate of 1.5%, and emulsified asphalt content of 2.9%. The measured value of gross volume density was 2.210 g/cm^3^.

The maximum theoretical density is the density reflecting aggregate gradation theory under certain conditions. Under certain conditions, it can reflect the density of the mixture itself, and can reflect the density of gradation. Maximum theoretical density can reflect the quality of gradation. Combined with gross volume density of the specimen, the void volume can be calculated. The theoretical maximum density was 2.51 g/cm^3^.

The void volume was 11.9%. It can be seen that the void volume meets the requirement of the specification, but it was too large, and so poorly resistant to water damage, which required the asphalt to have good bonding performance.

### 2.3. Specimens Preparation and Experimental Methods

The first step was to change the amount of cement and emulsified asphalt in the mixture, use the vibration molding method to design the mixture ratio, and decide the best mix ratio by Marshall test. The next step was to study the technical performance, including the volume characteristics of the mixture, water stability, and high-temperature stability of the Marshall test specimens made from the above type of asphalt mixture. The third step was to carry out fatigue tests of 2–4% cement content and 3–5% asphalt content test pieces under different stress conditions, analyze the factors affecting the fatigue performance of cold recycled mixtures, and compare the fatigue lives with those of other materials.

In this study, the vibration compaction method was used to form the specimens; the main working parameters of the vibratory compaction molding machine are as follows: the vibration frequency is 30 Hz, the amplitude is 0–25 mm, the static pressure is 2500 N, the exciting force is 9000 N, and the compaction function is adjusted by changing the vibratory compaction time. The gross volume density of the specimens was measured when they were compacted 50 times, cured at 60 for 40 h, compacted 25 times and cooled at room temperature for 12 h, and was measured using the underwater weighing method T0706-2000. The high-temperature stability was evaluated by a rutting test. The size of specimens used in the rutting test is 300 mm × 300 mm × 80 mm; the rutting test was conducted according to “asphalt mixture rutting test” (T0719-2011). The temperature of the test is controlled at (60 ± 0.5) °C and the tire pressure is (0.7 ± 0.05) MPa. The water stability of the specimens was evaluated by splitting strength test. The steps are as follows: after forming the specimens under the same conditions, the strength of the specimens are measured after natural curing for seven days. The fatigue properties of recycled mixtures are evaluated by indirect tensile fatigue test. According to the research results of SHRP, the fatigue failure above room temperature is mainly caused by the accumulation of deformation of materials, which has no significant fatigue significance. Therefore, SHRP suggests that fatigue failure above 20 °C should not be considered. The criterion of fatigue life is as follows: with an increase of loading times, the vertical deformation of specimens develops from stable stage to accelerated growth stage, and the number of loading times corresponding to the inflection point is defined as the number of fatigue failure effect [34]. Steps are given as follows: (1) Naturally cure the mixtures for three months after specimens have been formed whose size is φ150 × 150; and (2) Conduct a fatigue test on UTM-16 material testing machine at 15 °C by applying the sine-wave load with the frequency of 10 Hz. Five replicates for each condition were used.

## 3. Performance Evaluation of Emulsified Asphalt Cold Recycled Mixtures

### 3.1. High-Temperature Stability

High-temperature stability is one of the main indexes by which to evaluate the durability of asphalt mixtures. The heterogeneity of the asphalt mixture was determined by studying the movement characteristics of aggregate particles in asphalt mixture under load and temperature [35]. Rutting tests were conducted to reflect the resistance of cement cement-emulsified asphalt cold recycled mixtures to resist deformation at high-temperature.

For asphalt materials, the high-temperature stability of the materials used for road surface was the main consideration, while the temperature changes of the base layer and were relatively slow. At the same time, the emulsified asphalt content of the base layer was small, and the strength formation was mainly based on the strength of cement. The dynamic stability of the two gradations is shown in Table 5.

It can be seen that the dynamic stability of cold recycled mixture with 65% RAP is much higher than with 100% RAP, because of the incorporation of cement and the addition of emulsified asphalt. In addition, the technical requirement for dynamic stability of normal asphalt mixture is 800 cycles/mm according to the Chinese specification, and the dynamic stability of cement-emulsified asphalt mixture with 100% RAP can reach twice of the requirement. This means that the cement-emulsified asphalt mixture with 100% RAP could be applied as the asphalt layers in the pavements.

### 3.2. Water Stability

Water damage is one of the main problems of asphalt pavements. In general, the freeze-thaw splitting test in cold regions of the north can reflect water stability. For the rainy and warm climates in the southern areas, there is no freeze-thaw cycle, and the wet-dry splitting strength can be used as the control index of water stability. Studies have also found that [36,37] during the curing period, proper rainwater is beneficial for the strength of the mixture, because the hydration of cement cannot be separated from water. Proper rainwater contact is ensured in the early stage the curing period, and the actual road curing effect is better. Therefore, in this study, the strength of Marshall specimens was measured by natural curing for seven days under the same conditions. The freeze-thaw splitting strength and the wet-dry splitting strength are shown in Figure 4. Dry, freeze-thaw and wet splitting strengths of cold recycled emulsified asphalt mixtures are shown in Table 6.

### 3.3. Fatigue Property

Emulsified asphalt cold recycled mixtures are generally used for pavement bases, and fatigue damage is one of the main failure modes of pavement base structures. Therefore, fatigue properties are an important evaluation indicator of base materials [38]. In this study, fatigue tests of 2–4% cement content and 3–5% asphalt content test under different stresses conditions were carried out. The test results are shown in Table 7.

According to the results of previous studies, the relationship between fatigue number and stress ratio can be better reflected by fitting fatigue equations into a logarithmic form. Therefore, the relationship between the fatigue times and the stress is:(1)$$\frac{\sigma }{\sigma_{f}}=aN^{c}$$
where σ is the ultimate tensile strength of the recycled mixtures; $\sigma_{f}$ is Allowable tensile stress; and N is the -Fatigue life; *a* and *c*-are the regression coefficients.

This formula, taken the logarithm on both sides, can get $\log\left(\frac{\sigma }{\sigma_{f}}\right)=\log\left(aN^{c}\right)$, i.e., $\log\left(\frac{\sigma }{\sigma_{f}}\right)=\log a+c\log N$; that is to say, it can linearly regress into a linear equation y=c*x+m, such as $y=\log\left(\frac{\sigma }{\sigma_{f}}\right)$, m=loga, x=logN.

It can be seen from the fitting relationship in Figure 5 that the logarithm of the fatigue times of the wet and dry splitting fatigue and the logarithm of stress ratio satisfy the linear relationship. This can indicate that the number of fatigue times and the stress ratio fitted by this function are of practical value. The fatigue equations for different dosages are shown in Table 8.

## 4. Influencing Factors of Fatigue Properties for Emulsified Asphalt Cold Recycled Mixtures

### 4.1. Effect of Humidity on Fatigue Life of Specimens

This study selected a group with 4% asphalt content and 2% cement content and compared their wet-dry splitting fatigues. The result is shown in Figure 6.

From Figure 6, it can be seen that the number of dry fatigue times is more than that of wet fatigue times corresponding to the same stress, and the greater the stress ratio, the less obvious the difference of fatigue times. That is to say, for the cold recycled structural layer, the application of repeated loads under wet conditions should be avoided.

### 4.2. Effect of the Amount of Emulsified Asphalt on the Fatigue Life of Specimens

Figure 7 shows the fatigue times of three kinds of emulsified asphalts with 3%, 4%, and 5% asphalt contents and under different stress ratios with the same cement contents; the results show that the influence of emulsified asphalt content on the fatigue properties of specimens is complex. At low stress ratios, low emulsified asphalt dosages appear to be advantageous; but at high stress ratios, high emulsified asphalt dosages appear to be advantageous. In addition to the test errors, the main reason for the above test results is that the controlled stress fatigue mode was adopted in this test. This test method is not suitable for flexible materials, so it is insensitive to the reaction of asphalt dosage.

### 4.3. Effect of the Amount of Cement on the Fatigue Life of Specimens

Three emulsified asphalts with an asphalt content of 4% and cement contents of 1%, 2%, and 3% were selected for fatigue tests under different stress ratios. From the fatigue times under the corresponding stress ratio, it can be found that under a low stress ratio, the fatigue performance is better under the low cement content; while at the high stress ratios, the cement content has little effect on the fatigue times. The fatigue time of different cement content is shown in Figure 8.

### 4.4. Comparison with Other Materials

Because of the great difference between the indoor fatigue test and the actual fatigue process of pavements, it is difficult to evaluate the anti-fatigue performance simply by looking at the indoor fatigue test results. For this reason, this paper conducted a comparative study on emulsified asphalt cold recycled mixtures, foamed asphalt cold recycled mixtures, and hot asphalt mixtures. 

For foamed asphalt cold recycled mixtures, the same RAP materials and aggregate gradation as emulsified asphalt cold recycled mixtures were used. The asphalt was firstly foamed at 150 °C with 2.0% water consumption, the optimum foamed asphalt contents was determined as 2.0%, and the water content added in mixing process was determined as 4.8%. The aggregate gradation for hot asphalt mixtures is the same as that of temulsified asphalt cold recycled mixtures, and the optimal asphalt content is determined as 5.3%.

The fatigue life of cold recycled emulsified asphalt mixtures was compared with that of hot mix asphalt mixtures, and the relationship between stress ratio and fatigue times is shown in Figure 9 [39,40].

From Figure 9, it can be seen that the performance of emulsified asphalt cold recycled mixtures is not as good as that of the hot mix asphalt mixtures, in terms of fatigue times and high-low stress ratios. The strength of the hot mix asphalt mixtures is higher than that of the recycled material. Therefore, under the same stress ratio, hot mix asphalt mixtures tend to bear much more stress than the recycled material test specimens. Under the same stress level, the fatigue life of the hot mix asphalt mixtures is higher than that of the recycled materials, and the sensitivity of recycled material to stress increase is greater than that of hot mix asphalt.

Foamed bitumen is another way of cold recycling. In the process of cold recycling, the addition of active materials, such as cement, is also a major aspect of strength enhancement. According to the fatigue data of foam cold recycled asphalt, this study drew a curve and compared its fatigue characteristics with those of cement emulsified asphalt. The fatigue life of the cement emulsified asphalt cold recycled material was compared with the fatigue life of foam cold recycled asphalt mixtures. The relationship between the stress ratio and the fatigue number is shown in Figure 10.

It can be seen from Figure 10 that compared with the emulsified asphalt mixtures, the fatigue life of the foam cold recycled mixtures is almost the same as that of the emulsified asphalt mixtures. It can therefore be said that the fatigue degree of the two mixtures is equal.

## 5. Summary and Conclusions

Based on the analysis of the mechanism of cold recycled emulsified asphalt, a series of pavement performances, including composition design, high temperature stability, water stability, and fatigue characteristics of emulsified asphalt cold recycled mixtures, were studied in the laboratory by the vibration compaction molding method, which is more in line with real-world situations. The main conclusions are as follows:(1)The composition design of emulsified asphalt cold recycled mixtures was carried out by the vibration forming method. The results show that the performance of the mixture with 65% RAP meets the requirements. The proportion of the design grading is: RAP materials: 0–3 mm aggregates: 10–20 mm aggregates: mineral powder = 65:13:20:2, the optimal emulsified asphalt binder and cement dosages were determined as 2.9 % and 1.5% respectively.(2)The developed emulsified asphalt meets the technical requirements and can be used in the composition design of cold recycled mixtures. The well-designed emulsified asphalt cold recycled mixtures have good high-temperature stability and water stability, which can meet the application requirements for a highway base and underlying layer.(3)When the stress level is lower, the fatigue performance of the mixture with low emulsified asphalt dosage is better; when the stress level is higher, the dosage of high emulsified asphalt is more favorable; when the stress level is lower, the fatigue performance under low cement content is better; but when the stress level is high, the cement content has little effect on the fatigue frequency.(4)Emulsified asphalt cold recycled mixtures have poor fatigue resistance and lower fatigue life than hot mix asphalt mixtures. Under the same stress level, the fatigue life of the hot mix asphalt mixture is higher than that of the recycled materials, and the sensitivity of recycled material to stress increase is greater than that of hot mix asphalt. Therefore, in structural design, the cold recycled structure layer should, as far as possible, not be placed in the bending-tension fatigue zone.

## Figures and Tables

**Figure 1 materials-12-02682-f001:**
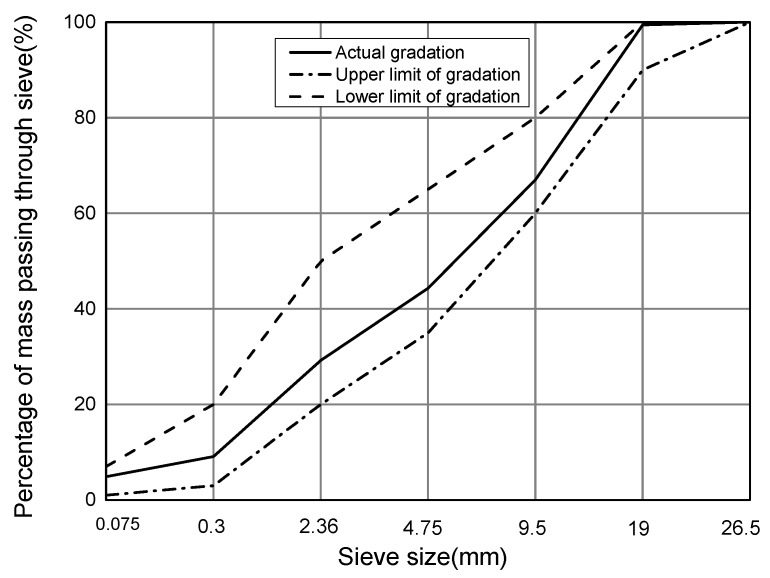
Gradation Curve of Emulsified Asphalt Cold Recycling Mixtures (65%RAP).

**Figure 2 materials-12-02682-f002:**
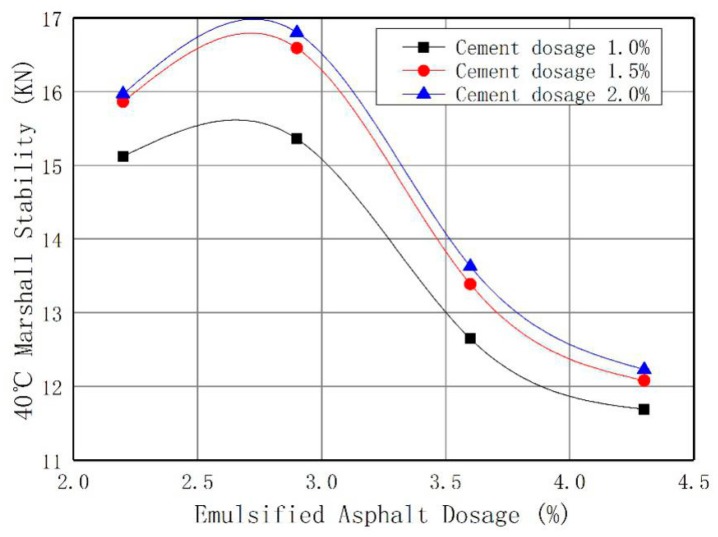
Marshall Stability Trend Chart.

**Figure 3 materials-12-02682-f003:**
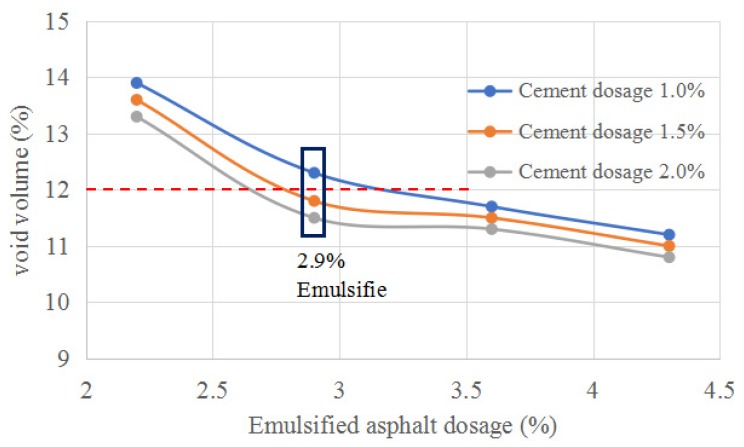
Void volume Trend Chart.

**Figure 4 materials-12-02682-f004:**
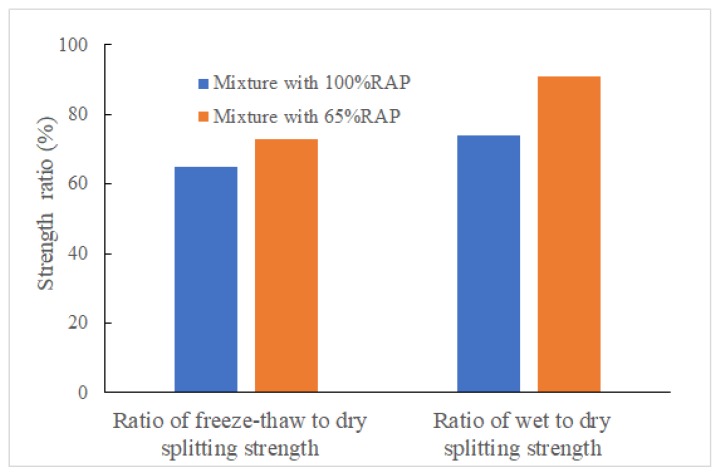
Comparison of Strength of Two Mixtures.

**Figure 5 materials-12-02682-f005:**
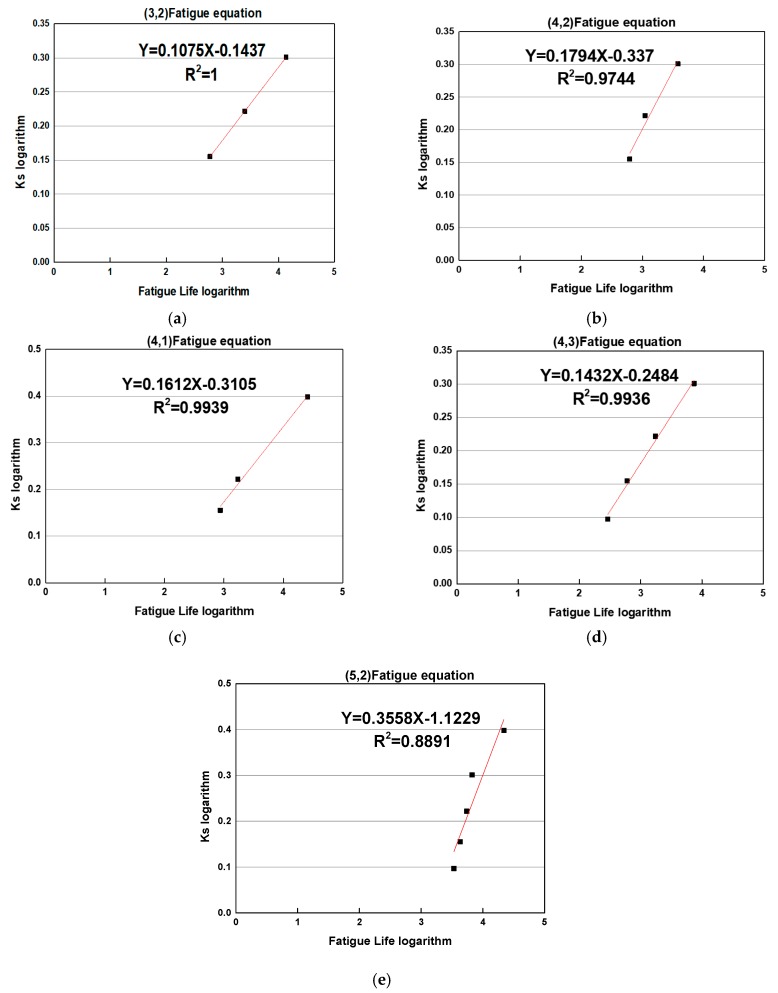
Logarithm of fatigue life with different parameters. (**a**) (3,2) Fatigue equation; (**b**) (4,2) fatigue equation; (**c**) (4,1) fatigue equation; (**d**) (4,3) fatigue equation; (**e**) (5,2) fatigue equation.

**Figure 6 materials-12-02682-f006:**
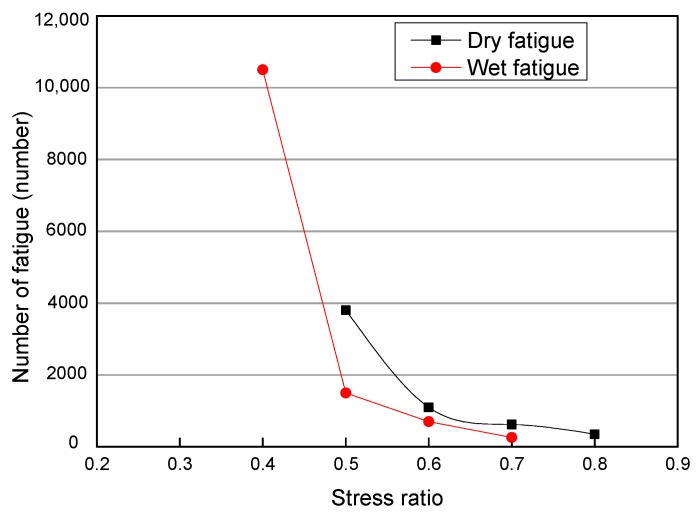
Comparison of dry and wet fatigue times.

**Figure 7 materials-12-02682-f007:**
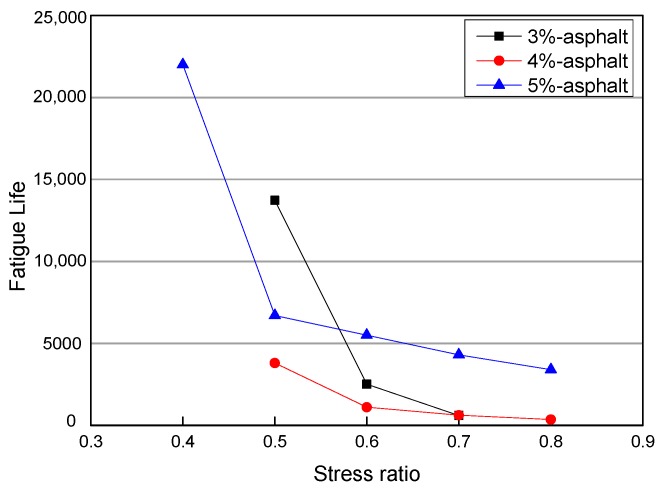
Fatigue times of different asphalt contents.

**Figure 8 materials-12-02682-f008:**
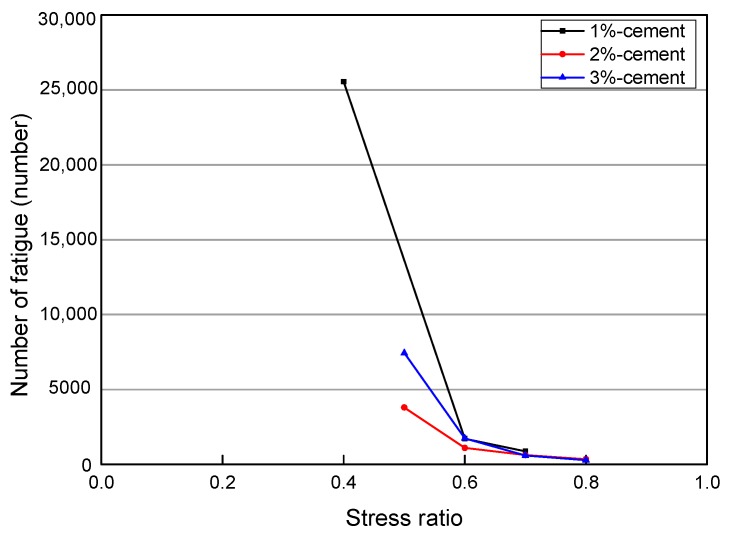
Fatigue times of different cement contents.

**Figure 9 materials-12-02682-f009:**
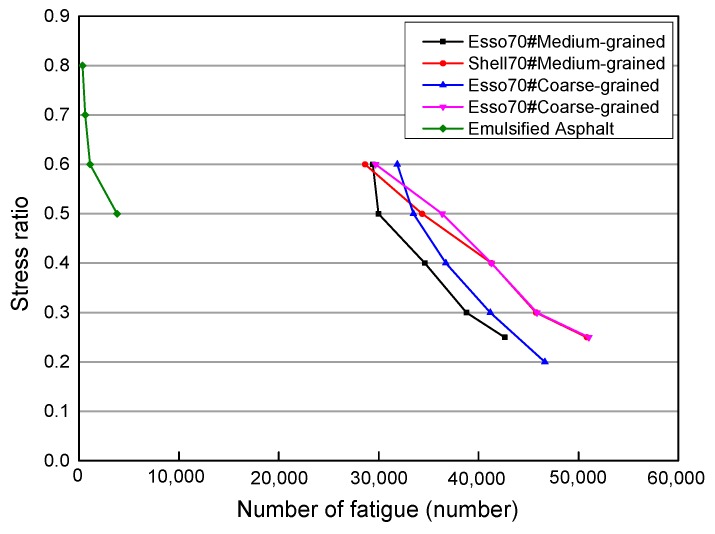
Comparison of Emulsified Asphalt Cold Recycled Mixtures and Hot Mix Asphalt Mixtures.

**Figure 10 materials-12-02682-f010:**
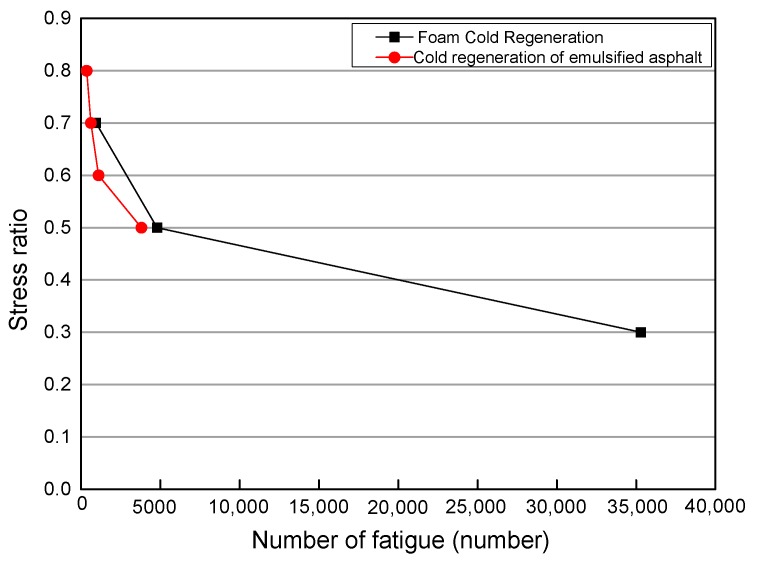
Comparison of emulsified asphalt cold recycled mixtures and foamed asphalt mixtures.

**Table 1 materials-12-02682-t001:** Technical properties of emulsified asphalt.

Indexes	Test Results	Technical Requirements	Technical Method
Screen residual (1.18 mm), %	<0.1	≤0.1	T0652
Determination of Residue by Evaporation, %	58.7	≥55	T0651
Demulsification Rate	Slow	Slow	T0658
Charge	Cationic	Cationic	T0653
Penetration (25 °C), 1/10mm	88	45–150	T0604
Softening Point, °C		-	T0606
Ductility (15 °C), cm	87.0	≥40	T0605
Storage Stability 5d	3.5	<5	T0655
Solubility (Trichloroethylene), %	99	>97.5	T0607
Standard Viscosity C25.3, s	15.0	10–60	T0621
binding area	≥4/5	≥4/5	

**Table 2 materials-12-02682-t002:** Technical properties of P.O. 32.5 cement.

Indexes		Test Result	Technical Requirements
Fineness (%) (0.08 mm square hole sieve residue)		3.4	≤15
Initial setting time (min)		180	≥45
Final setting time (min)		360	≤390
Stability (boiling)		Qualified	Qualified required
Compressive strength (MPa)	3 days	20.1	≥16
	28 days	48.8	≥42.5
Flexural strength (MPa)	3 days	4.6	≥3.4
	28 days	7.8	≥6.4

**Table 3 materials-12-02682-t003:** Passing rate of the key sieve for the design mixture.

**Sieve size/mm**	26.5	19	9.5	4.75	2.36	0.3	0.075
**Passing rate/%**	100	99.4	67.0	44.3	29.2	9.1	4.9

**Table 4 materials-12-02682-t004:** Splitting test results of emulsified asphalt cold recycled mixtures.

Cement Content (%)	Emulsified Asphalt Content (%)	15 °C Splitting Strength (MPa)	Technical Requirements (MPa)
1.0	2.9	0.56	≥0.5
1.5	2.9	0.59	
2.0	2.9	0.61	

**Table 5 materials-12-02682-t005:** Dynamic stability of two gradations.

Grading Type	45 min Deformation/mm	60 min Deformation/mm	Dynamic Stability/Cycles·mm^−1^
100% RAP	3.683	4.087	1560
65% RAP	3.293	3.535	2157

**Table 6 materials-12-02682-t006:** The dry, freeze-thaw and wet splitting strengths.

Mixtures Type	Dry Splitting Strength (MPa)	Freeze-Thaw Splitting Strength (MPa)	Wet Splitting Strength (MPa)
Mixture with 100%RAP	0.23	0.15	0.17
Mixture with 65%RAP	0.45	0.33	0.41

**Table 7 materials-12-02682-t007:** Fatigue times of recycled materials with different parameters.

Asphalt Content (%)	Cement Content (%)	Splitting Strength (MPa)	Stress Ratio	Dry Splitting Fatigue Life	Wet Splitting Fatigue Life
3	2	0.65	0.5	13,725	
			0.6	2512	
			0.7	600	
4	2	0.71	0.4		10,500
			0.5	3800	1500
			0.6	1100	700
			0.7	620	260
			0.8	350	
5	2	0.62	0.4	22,000	
			0.5	6700	
			0.6	5500	
			0.7	4300	
			0.8	3400	
4	1	0.59	0.4	25,550	
4	3	0.80	0.5	7438	
			0.6	1736	
			0.7	600	
			0.8	290	

**Table 8 materials-12-02682-t008:** Fatigue equations for different asphalt and cement contents.

Emulsified Asphalt and Cement Content	Linear Equation	Fatigue Equation
(3%, 2%)	y = 0.1075x − 0.1437	$\log\frac{\sigma }{\sigma_{f}}=0.1075\log N-0.1437$
(4%, 2%)	y = 0.1794x − 0.337	$\log\frac{\sigma }{\sigma_{f}}=0.1794\log N-0.337$
(5%, 2%)	y = 0.3558x − 1.1229	$\log\frac{\sigma }{\sigma_{f}}=0.3558\log N-1.1229$
(4%, 1%)	y = 0.1612x − 0.3105	$\log\frac{\sigma }{\sigma_{f}}=0.1612\log N-0.3105$
(4%, 3%)	y = 0.1432x − 0.2484	$\log\frac{\sigma }{\sigma_{f}}=0.1432\log N-0.2484$

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
