# Peer review of "Composition Design and Performance Evaluation of Emulsified Asphalt Cold Recycled Mixtures"

_materials, 2019, doi:10.3390/ma12172682_

Round 1
Reviewer 1 Report
Unfortunately, there is little information presented in this article that is not well known by those working in the industry. It is a well known fact that current mix design practices do not evaluate fatigue life and therefore, Wirtgen and the Asphalt Recycling & Reclaiming Association both recommend limits on the amount of cement to protect from a brittle mix. This is done as adding fatigue testing to a mix design is impractical and cost prohibitive. Wirtgen recommends a maximum of 1% cement and ARRA recommends the minimum ratio of residual asphalt to cement of 2.5:1 or 3:1. The cement contents in this study were above the industry recommended amounts. The results were as expected with little new information.
Author Response
Dear Reviewer:
Thank you very much for reviewing our manuscript entitled “Composition design and performance evaluation of emulsified asphalt cold recycled mixtures” (Manuscript ID: materials-542209). Your comments are all valuable and helpful for revising and improving our manuscript, as well as the important guiding significance to our research.
We have studied comments carefully and made correction which we hope meet with approval. The revised portion are highlighted in the revised version.

Reviewer 2 Report
Please correct your emulsification procedure because you state:"The emulsification process adopted in this paper was as follows: At First, water, emulsifier and additive were mixed into solution at 60℃. Then, the solution was mixed with asphalt of 70 # at 130℃ and added into the emulsifier to obtain asphalt emulsion."
Thus you are adding emulsifier with a mixture and then adding this to asphalt and then another emulsifier or what.
You should explain more the behavior of stress and why it is following such trend.
Author Response

(The authors gave the same response as above.)

Reviewer 3 Report
All authors in this manuscript have made great efforts to describe the investigation of the emulsified asphalt binders and cement as the cold recycled mixtures in terms of high-temperature stability, water stability as well as the fatigue characteristics. Despite authors’ great efforts, there are a few specific issues the authors had better be addressed by making modifications to the manuscript or by clarifying in their response, after which I would consider this work suitable for publication and readers.
At line 99-100 in page 3, the aggregate gradation was mentioned about 65% RAP which consisted of 13% new aggregates (0~3mm), 20% new aggregates (10~20mm) and 2% mineral powders. And the gradation results are shown in Table 3. What is the meaning of 13%, 20%, and 2% of new aggregates? In order to clearly understand the gradation results, please plot the gradation of aggregate size as well.
In page 4, how did all authors decide the amount of emulsified asphalt?
In page 5, authors mentioned that the optimum emulsified asphalt content and cement content is 2.9% and 1.5%, respectively. However, under 2.9% of emulsified asphalt content, the cement dosage 2.0% can be satisfied in void requirement 10%, instead of cement dosage, 1.5% as shown in Figure 2. Could you explain why 1.5% of cement content is optimum ?
In page 6, What is the abbreviation of SHRP?
At line 189 in page 6, the dynamic stability of cement-emulsified asphalt is much higher than that of old asphalt mixture which can reach twice of that of the old asphalt mixture. But, based on dynamic stability in Table 5, I can not understand the meaning of twice. Could you explain of it?
In Figure 3, could you note the meaning of grading type in Figure 3?
The legend of Figure 6, 7 and 8 should be marked more clearly and largely.
Author Response

(The authors gave the same response as above.)

Round 2
Reviewer 1 Report
Revisions are acceptable